# Is the Proteome of Bronchoalveolar Lavage Extracellular Vesicles a Marker of Advanced Lung Cancer?

**DOI:** 10.3390/cancers12113450

**Published:** 2020-11-20

**Authors:** Ana Sofia Carvalho, Maria Carolina Strano Moraes, Chan Hyun Na, Ivo Fierro-Monti, Andreia Henriques, Sara Zahedi, Cristian Bodo, Erin M Tranfield, Ana Laura Sousa, Ana Farinho, Luís Vaz Rodrigues, Paula Pinto, Cristina Bárbara, Leonor Mota, Tiago Tavares de Abreu, Júlio Semedo, Susana Seixas, Prashant Kumar, Bruno Costa-Silva, Akhilesh Pandey, Rune Matthiesen

**Affiliations:** 1Computational and Experimental Biology Group, Chronic Diseases Research Centre, NOVA Medical School, Faculdade de Ciencias Medicas, Universidade NOVA de Lisboa, Campo dos Martires da Patria, 130, 1169-056 Lisboa, Portugal; ivo.fierromonti@nms.unl.pt (I.F.-M.); andreia.henriques@nms.unl.pt (A.H.); sara.zahedi@nms.unl.pt (S.Z.); 2Systems Oncology Group, Champalimaud Research, Champalimaud Centre for the Unknown, Av. Brasilia, Doca de Pedroucos, 1400-038 Lisbon, Portugal; carolina.stranomoraes@research.fchampalimaud.org (M.C.S.M.); cristian.bodo@research.fchampalimaud.org (C.B.); bruno.costadasilva@research.fchampalimaud.org (B.C.-S.); 3Department of Neurology, Institute for Cell Engineering, Johns Hopkins University School of Medicine, Baltimore, MD 21205, USA; chanhyun@jhmi.edu; 4Electron Microscopy Facility, Instituto Gulbenkian de Ciência—Rua da Quinta Grande, 6, 2780-156 Oeiras, Portugal; etranfield@igc.gulbenkian.pt (E.M.T.); alsousa@igc.gulbenkian.pt (A.L.S.); 5iNOVA4Health—Advancing Precision Medicine, CEDOC—Chronic Diseases Research Centre, NOVA Medical School/Faculdade de Ciências Médicas, Universidade NOVA de Lisboa, Campo dos Martires da Patria, 130, 1169-056 Lisboa, Portugal; ana.farinho@nms.unl.pt; 6Department of Pneumology, Unidade Local de Saúde da Guarda (USLGuarda), 6300-659 Guarda, Portugal; luis.vaz.rodrigues@gmail.com; 7Unidade de Técnicas Invasivas Pneumológicas, Pneumologia II, Hospital Pulido Valente, Centro Hospitalar Lisboa Norte, 1649-028 Lisbon, Portugal; paulagpinto@gmail.com (P.P.); 37749@chln.min-saude.pt (L.M.); tavaresdeabreu@gmail.com (T.T.d.A.); jasemedo@netcabo.pt (J.S.); 8Instituto de Saúde Ambiental (ISAMB), Faculdade de Medicina, Universidade de Lisboa, Centro Hospitalar Universitário Lisboa Norte, 1649-028 Lisbon, Portugal; cristina.barbara@chln.min-saude.pt; 9Instituto de Investigação e Inovação em Saúde (I3S), Universidade do Porto, 4200-135 Porto, Portugal; sseixas@ipatimup.pt; 10Institute of Bioinformatics, Discoverer building, ITPL, Bangalore 560066, India; prashant@ibioinformatics.org (P.K.); pandey.akhilesh@mayo.edu (A.P.); 11Manipal Academy of Higher Education (MAHE), Manipal 576104, India; 12Department of Laboratory Medicine and Pathology, Mayo Clinic, Rochester, MN 55905, USA

**Keywords:** extracellular vesicles, lung cancer, proteomics, immuno oncology, bronchoalveolar lavage

## Abstract

**Simple Summary:**

Bronchoalveolar lavage is routinely collected during bronchoscopy for cytology analysis in the diagnostic of lung cancer. Due to low sensitivity of this method, early-stage cancers are undetected, lowering the treatment success. In this study, we analyzed extracellular vesicles isolated from bronchoalveolar lavage of lung cancer suspects by mass spectrometry-based proteomics. The protein composition of bronchoalveolar lavage extracellular vesicles of late-stage cancer showed a higher proteome complexity associated with mortality within the two year follow-up period. We identified a potential therapeutic target DNMT3B complex which was significantly expressed in bronchoalveolar lavage extracellular vesicles as well as in tumor tissue. Bronchoalveolar lavage extracellular vesicles proteome analysis of immune markers indicates the presence of markers of innate immune and fibroblast cells.

**Abstract:**

Acellular bronchoalveolar lavage (BAL) proteomics can partially separate lung cancer from non-lung cancer patients based on principal component analysis and multivariate analysis. Furthermore, the variance in the proteomics data sets is correlated mainly with lung cancer status and, to a lesser extent, smoking status and gender. Despite these advances BAL small and large extracellular vehicles (EVs) proteomes reveal aberrant protein expression in paracrine signaling mechanisms in cancer initiation and progression. We consequently present a case-control study of 24 bronchoalveolar lavage extracellular vesicle samples which were analyzed by state-of-the-art liquid chromatography-mass spectrometry (LC-MS). We obtained evidence that BAL EVs proteome complexity correlated with lung cancer stage 4 and mortality within two years´ follow-up (*p* value = 0.006). The potential therapeutic target DNMT3B complex is significantly up-regulated in tumor tissue and BAL EVs. The computational analysis of the immune and fibroblast cell markers in EVs suggests that patients who deceased within the follow-up period display higher marker expression indicative of innate immune and fibroblast cells (four out of five cases). This study provides insights into the proteome content of BAL EVs and their correlation to clinical outcomes.

## 1. Introduction

At present, lung cancer persisted as the most prevalent oncological disease with an estimated incidence of more than 2 million new cases worldwide, and a mortality rate of 84% with a total of 1.8 million deaths yearly [1,2]. The late detection of the disease is the main cause of such a dismal outcome as demonstrated by the five-year overall survival rates for non–small cell lung cancer (NSCLC): 14–49% for stages I to IIIA, and <5% for stage IIIB/IV [3].

The characterization of lung tumor tissue and the surrounding microenvironment at the molecular level contributed to the increased knowledge on the physiology of the disease as well as the design of novel therapies. Nevertheless, the collection of tissue biopsies of peripheral adenocarcinomas in distal airways by minimally invasive techniques is limited, and during the past years, several studies have targeted different types of biological specimens, which range from tumor tissues to different types of liquid biopsies [4]. Bronchoalveolar lavage (BAL) liquid biopsies are obtained from minimally invasive procedures, for example, fiber optic bronchoscopy. Bronchoscopy is typically performed upon suspicion of lung cancer, for example, derived from previously obtained imaging of the lungs. Due to the closer anatomical proximity of BAL compared to saliva, more accurate biomarker signatures are expected from BAL than from biofluids such as blood, saliva, or sputum. Blood and saliva, on the other hand, might serve as potential markers for early screening. In the past, several molecular technologies were applied to profile the molecular content of BAL such as miRNA [5], mRNA [6], DNA [7], DNA methylation [8], metabolites [9], microbiota [10], and proteomics. Over the years, proteomics technologies have evolved and consequently, BAL proteome profiling has been attempted multiple times. The first BAL proteomics profiling applied MALDI TOF-MS [11]. Later, 2D-PAGE and MALDI-TOF were combined [12]. Nowadays, liquid chromatography-mass spectrometry (LC-MS) approaches are typically applied for BAL proteome profiling [13,14,15]. Abundant proteins like albumin partially hamper the protein coverage obtainable by direct LC-MS analysis of acellular BAL samples. Therefore, Sim et al. [16] established a novel methodology based on combining antibody-based depletion of high abundant BALF proteins, high pH peptide fractionation, and label-free quantitation on a high-resolution Orbitrap Fusion Lumos instrument. However, antibody depletion of abundant proteins risks the removal of clinically relevant target proteins. Consequently, it is imperative to explore additional sample preparation methodologies.

The enrichment of small (sEVs) and large (lEVs) EVs represents an alternative methodology for the depletion of abundant proteins in biomarker studies based on liquid biopsies [17]. Currently, no study has targeted BAL sEVs and lEVs from suspected lung cancer patients. We, therefore, performed an exploratory case-control study to evaluate the protein complexity from proteomics profiling of BAL EV fractions. The sEVs proteins constituted the most complex proteome when compared to acellular BAL, vesicle-depleted BAL, and lEVs. We identified and quantified 7484 protein isoforms from 3158 encoding genes from sEVs. The lEVs and sEVs were enriched by sequential centrifugation followed by ultracentrifugation and floatation on sucrose gradient cushion, which captures all the subpopulations of sEVs while depleting from non-vesicular protein aggregates and complexes. Besides, the potential of BAL sEV proteome as a source of biomarkers was explored. The data obtained suggested that BAL sEV proteome complexity correlates with cancer stage IV and death (Wilcoxon rank-sum test *p* value = 0.006). The proteomes were compared, by system biology approaches, to previously obtained proteomes from lung tumor tissues and acellular BAL proteome [13,18]. sEVs and tumor tissue displayed a common significant regulation of several enriched functional categories with potential therapeutic value. Finally, sEV protein markers for innate immune and fibroblast cells correlated with poor prognosis (mortality within two years follow-up in four out of five cases).

## 2. Results

### 2.1. Outline of Study

As a proof of concept, we compared different BAL fractions by high-throughput mass spectrometry analysis such as acellular BAL, vesicle depleted BAL, lEVs and sEVs from control, and lung cancer patients. The proteome content was compared by LC-MS (Appendix A). We observed that sEVs contained the larger proteome complexity with the most significant potential for biomarker discovery (Section 2.2 and Appendix A). Therefore, sEVs were isolated from 24 acellular BAL samples and analyzed by mass spectrometry and further characterized by transmission electron microscopy (TEM), western blot, and nanoparticle tracking analysis. The quantitative MS data were compared with iBAQ values obtained from reanalyzed data of previous studies on acellular BAL (PXD004700) [13] and patient-matched tumor versus normal tissue (PXD000853) [18].

### 2.2. Proteome Content of lEVs, sEVs, Acellular BAL, and EV Depleted BAL (DB)

BAL fluids (acellular BAL, aBAL) from two different cancer status samples (NO and YES) were fractionated by sequential centrifugation and ultracentrifugation into three fractions: (1) depleted BAL (DB), (2) lEVs and (3) sEVs (Appendix A). To address which BAL fraction holds the greatest promise in terms of significant cancer status discrimination based on protein identification and quantitation, each fraction was analyzed by LC-MS in parallel with the respective aBAL fluid. Figure 1 compares the protein identification in cancer versus control for (a) sEVs, (b) BAL, (c) lEVs, and (d) DB. The sEVs fraction resulted in the highest number of protein identifications and the highest number of proteins unique to cancer (Figure 1a). KEGG functional enrichment of all the identifications from each of the fractions was compared to address the question of which fraction has the greatest potential for separating cancer status (Figure 1e). Appendix A displays the overlap between the identified proteins in each fraction type obtained from LC-MS analysis. The highest numbers of quantified proteins were obtained in BAL sEVs from YES and NO cancer status samples. On average the number of identified proteins from sEVs was almost double compared to aBAL, while in BAL vesicle-depleted and lEVs fractions the number of identified proteins was lower, with the lEVs proteome being the simplest proteome (Appendix A). KEGG functional enrichment analysis of all identified proteins revealed functional diversity between the BAL fractions analyzed (Figure 1e). The samples clustered according to BAL fractions based on the functional KEGG annotation, strongly suggesting that each fraction contains distinct protein functional groups. The identified proteins overall exhibited significant enrichment in pathways related to infection, inflammation, and complement and coagulation pathways. sEVs uniquely displayed significant enrichment in proteasome and ribosome factors while no enrichment in glycolysis and glycogenesis factors was observed in contrast to the other BAL fractions (Figure 1). Proteasome and ribosome factors are known as abundant factors in cancer due to increased protein turnover. Although lEV fractions resulted in the lowest amount of identifications, it presented unique functional enrichment related to, e.g., chemokine and synapse signaling in cancer. In turn, the two vesicles containing fractions displayed the maximum separation in terms of sample clustering. None of the fractions fully encapsulated all enriched functional categories. Proteomics analysis of sEV fractions showed the highest number of identified proteins and the highest diversity of functional enriched KEGG pathways. In this view, we have analyzed the proteome of sEVs for biomarkers discovery in a case-control study of 12 controls and 12 lung cancer cases.

### 2.3. Baseline Characteristic of Clinical Samples for the Enrichment of sEVs

To explore the potential of sEVs proteome ability to classify clinical samples, we selected 12 controls (non-lung cancer cases) and 12 lung cancer samples in a case-control design (Table 1) from our previous analyzed cohort of 91 patients suspected of lung cancer [13]. The lung cancer status reflects the diagnosis after two years of follow-up. Approximately 42% of the cancer patients died within the two year follow-up period. The selection of samples for sEV characterization was optimized to prevent statistical association of age, gender, smoking status, and experimental batch effects with lung cancer status. As expected, the lung cancer status displayed an association to two-year survival and cancer staging. sEVs from the 24 clinical samples were isolated following the protocol outlined in (Appendix A).

### 2.4. Nanoparticle Tracking Analysis of Isolated EVs

The enriched sEVs were characterized by nanoparticle tracking analysis (Figure 2 and Appendix A). Figure 2 displays representative particle size distribution for one control (a) and one lung cancer sample (b). All the normalized particle distributions are plotted in Appendix A and color-coded with clinical status. The distributions display high similarity across samples. Based on a t-test on the normalized counts for each particle size bin of 0.5 nm, no statistical differences between the distributions for control versus cancer were observed (Appendix A). All distributions displayed a maximum peak around 100 nm, which represents the expected size mode of sEVs. The precise position of the maximum peak differed slightly between samples. The different samples displayed slightly different peak tops at larger particle sizes likely representing different types of non-vesicle particles such as protein complexes, aggregates, and eventually virus particles. A comparison of overall EV protein content and overall total particle counts from control versus cancer samples showed no significant differences (Figure 2c,d).

### 2.5. Transmission Electron Microscopy and Overall Proteome of EVs

In accordance with nanoparticle tracking analysis, TEM analysis of sEVs revealed vesicles of similar sizes around 100 nm (Figure 3a). Western blot analysis of the exosome marker CD63 comparing BAL to sEVs fraction showed a fortyfold enrichment (Appendix A). All identified proteins from controls and lung cancer in sEV were compared to proteins deposited in the exosome database ExoCarta, showing an overlap >70% for the two different cancer status samples (Figure 3b). However, such comparisons exclude protein quantitation information and consequently are not suited for assessing the exosome content in the sEV fractions. We, therefore, developed a bioinformatics methodology to quantitatively estimate exosome content in sEV fractions (Figure 3c). The heatmap displays average iBAQ values ranked from 0 to 1 for frequently reported exosome markers in the literature and the 10 most frequently reported proteins in ExoCarta. The quantitative values from cell line sEVs were obtained by re-analyzing previously published MS data and serve as reference samples. The reference data for sEVs from cell lines were obtained by two different EV enrichment methodologies: (1) PEG-based precipitation [19] and (2) differential ultra-centrifugation [20]. The two methods resulted in the identification and quantification of similar exosome markers. We compared the quantitative proteomics data on sEVs from human clinical samples to the reference data on sEVs from cell lines. This constitutes a prudent MS-based assessment of exosome content. The clinical BAL sEVs isolated in this study exhibited similar expression levels of exosome markers when compared to the previous data from sEVs isolated from cell lines (Figure 3c) [19,20]. Additionally, the level of non EV proteins is considerably lower in BAL sEVs compared to cell lines sEVs enriched using the two methods while PEG isolated EVs from cell lines displayed the highest level of non EV protein markers.

### 2.6. Dysregulated Proteins in BAL Exosomes

Principal component analysis of all the quantitative values separated NO versus YES cancer samples based on three different principal components (Appendix A). The statistical analysis of iBAQ expression values by the R package limma applying a correction for gender and smoking resulted in 166 genes significantly regulated (*p*-value < 0.05) and one protein after correction of multiple testing (adjusted *p*-value < 0.05) between cancer positive and negative status (Appendix A and Figure 4a,b). Previously, we reported 133 significant regulated proteins after correction of multiple testing [13], in which 91 acellular BAL samples were analyzed. We speculate that this difference primarily results from the difference in the number of cases studied (24 versus 91) rather than reflecting BAL sEVs potential as a biomarker source. In other words, due to cancer heterogeneity, a higher number of samples are required to define proteins significantly dysregulated after correction of multiple testing. To further explore protein heterogeneity of sEVs samples, uniquely identified proteins in each sample were extracted. Figure 4c depicts the number of unique proteins, finding 607 unique proteins (Appendix A) in total for sEVs from lung cancer versus 176 for control. Counting the number of uniquely identified proteins per sample revealed that stage IV cancer and mortality displayed a significant tendency to contain more unique proteins identified compared to other cases (*p*-value = 0.006, Figure 4c). The *p*-value was calculated based on Wilcoxon rank-sum test without removing any outliers. This trend of increased protein complexity in cancer was confirmed in lung cancer tissue (Appendix A) but not for acellular BAL proteome (Appendix A). Unique cancer sEVs proteins displayed KEGG functional regulation in pathways associated with cancer (Figure 4d).

### 2.7. Complete Functional Regulation

Complete functional regulation analysis tests for significant regulation of entities (genes, proteins) within a functional group together with significant enrichment in terms of the identification of a given functional annotated group [21]. The methodology combines functional regulation and enrichment analysis into a single visual summary. Figure 5 compares the results from the complete functional regulation analysis of acellular BAL, sEVs, lung tumor tissue, and mouse xenotransplant lung tumor tissue. For each functional group, the number of proteins identified in each sample type is displayed as well as the total number of proteins in the respective category. A statistical test is performed by assessing the significance of the enrichment and the significance of the overall regulation of the category. Complete functional regulation analysis revealed higher similarity between sEVs and tumor tissue when compared to acellular BAL (Figure 5). The large functional groups up-regulated in both tissue and sEVs include NADP binding and ERK pathway. ECM receptor interaction constitutes the largest down-regulated functional category common to tissue and sEVs (Figure 5). Several functional categories revealed reverse regulation when comparing sEVs with tumor tissue, for example, triglyceride metabolic processes. DNMT3B complex showed up-regulation in lung cancer for sEVs and tumor tissue. Aberrant DNA methylation caused by the DNMT3B complex in lung cancer is well established and proposed as a possible therapeutic target.

### 2.8. Quantitative Evaluation of Immune Cell Lineage Markers

BAL contains cells from various sources like tumor cells, epithelial cells, immune cells, and fibroblasts [22,23]. Based on the R package MCPcounter [24], we estimated the average abundance scores of markers for eight different immune and two stromal cell populations (Figure 6). MCPcounter identified six protein markers in BAL, 15 in sEVs, and 17 in tissue samples. In BAL sEVs, B lineage markers resulted in the highest average abundance scores. BAL sEVs from lung cancer tend to display higher abundance scores for monocytic lineage and fibroblasts than controls (Figure 6). This trend was not confirmed for acellular BAL proteome (Appendix A). Markers of eight immune and two stromal cell populations could separate samples into normal, tumor, and xenotransplant tumor tissues (Appendix A).

## 3. Discussion

Deep protein coverage is paramount for identifying cancer signatures as well as contributing to the design of novel therapies [4]. Recent methodologies combining antibody-based depletion of high abundant BALF proteins, high pH peptide fractionation, and label-free quantitation on a high-resolution Orbitrap Fusion Lumos instrument demonstrated superior protein coverage [16]. In this study, we demonstrated the enrichment of sEVs by the sequential centrifugation method and quantification, resulted in a total of 7484 protein isoforms. The sEVs protocol typically yields sEVs fractions containing more than 90% vesicles according to flow cytometry experiments. Furthermore, the quality of the sEVs was accessed by nanoparticle tracking analysis, transmission electron microscopy, LC-MS, and western blotting. Webber and Clayton [25] argued that ratios >3 × 10^10^ particles/microgram of protein corresponded to high vesicular purity whereas ratios of 2 × 10^9^ to 2 × 10^10^ particles/microgram of protein indicated low purity. Webber and Clayton achieved solely ratios >3 × 10^10^ particles/microgram of protein from EVs isolated from cell culture conditioned media. On the other hand, Webber and Clayton [25] described for the body fluids urine and serum a 10-fold lower ratio similar to that obtained BAL for sEVs. In Figure 3c, we observe that the level of BCL2 (mitochondrial marker), GOLGA2 (Golgi marker), NUP98 (nuclear marker), and CANX (endoplasmatic reticulum marker) is much lower in our clinical sEVs isolates than what was obtained from cell lines from previously published studies. Based on our observation, we speculate that EVs from body fluids bind more extracellular proteins than EVs from cell culture conditioned media thereby explaining the lower particles to protein ratio for clinical derived EVs. The identified proteins in aBAL and BAL sEVs displayed common functional KEGG pathways enriched in infection, inflammation, and complement and coagulation cascades (Figure 1). Glycolysis/gluconeogenesis, together with amino acid metabolism, was enriched for BAL but completely depleted for sEVs. In contrast, “antigen processing and presentation” and proteins involved in virus infection were enriched preferentially in sEVs rather than BAL. In fact, many HLA variants were identified in BAL sEVs. HLA peptidome has recently gained interest as a source of neoantigen and biomarkers [26]. Our findings reinforce the novel methodologies for enriching for HLA for MS-based peptidome studies. The soluble HLA has been reported as a biomarker in various studies [27]. However, according to our data, neither aBAL nor depleted BAL showed enrichment for HLA suggesting that soluble HLA is mainly associated with sEVs.

Proteasome and ribosome factors were more enriched in sEVs than aBAL. These protein factors may co-purify during EVs enrichment based on densitometry. A recent study based on biotinylation of external proteins demonstrated that proteasome is located externally in EVs [28].

Here, we proposed a novel MS-based bioinformatics assessment of sEV for exosome markers. We further showed that the complete profiling of previously proposed exosome markers demonstrated that BAL sEVs contain a similar abundance of exosome markers compare to sEVs from cell lines. This is an efficient method compared to western blot (Figure 3c). We used 8 mL of acellular BAL for which we typically obtained 20 to 80 micrograms of sEV proteins. This limits the number of western blots that can be performed, as MS-based proteomics (approximately 2 micrograms) requires less protein compared to western blotting techniques (>5 micrograms).

Notably, we observed less significant regulated proteins after correction of multiple testing compared to our previous study on acellular BAL proteome [13]. We ascribe this to tumor heterogeneity, which means that larger cohorts are needed to define statically significant patterns in tumor sEVs. The unique proteins per sample supported this view (Figure 4b) and suggested that protein complexity in BAL sEVs and tissue but not acellular BAL correlates with tumor burden (Figure 4b and Appendix A). The follow-up studies on larger cohorts are needed to validate this observation. Furthermore, clinical utility must be validated in larger cohorts.

The complete functional regulation and enrichment analysis demonstrated that the functional regulation in sEV proteome revealed higher similarity to tumor tissue compared to acellular BAL. Similar results were obtained when comparing 60 NCI cell lines cellular proteome versus EV proteome using functional analysis [17]. The largest functional category in terms of identified proteins and significantly up-regulated in both BAL EVs and tissue were proteins involved in ERK pathway activation. This pathway is well described as aberrantly activated in cancer [29], including lung cancer [30,31]. Clinical targeting has been unsuccessful due to drug resistance and toxicity [32]. This highlights the emerging role of extracellular vesicle-mediated drug resistance in cancers ascribed to intercellular transfer of drug-resistant traits such as proteins or nucleic acids [33]. Efforts to elucidate the role of EVs in drug resistance implicated in downstream effectors of EGFR signaling cascade involving RAS-RAF-MEK-ERK pathways are therefore much needed. For example, it is conceivable that EGFR in EVs affects drug resistance by a similar mechanism as previously described in which CD20 targets, excessively secreted in EVs, shield the CD20 cellular target on cancer cells [34]. DNMT3B (Cytosine-5-methyltransferase 3β) complex proteins were also up-regulated in sEVs and tumor tissue. DNMT3B is an epigenetic modifier and a downstream substrate of EGFR signaling cascade via AKT pathway axis. Previous studies found that cancer cells express aberrant DNMT3B transcripts encoding truncated proteins which are hypermethylated, and transcriptionally inactivate tumor suppressor genes [35]. Various solid cancers overexpress DNMT3B transcripts including lung cancer, and DNMT3B is considered a potential therapeutic target [36]. This raises the question of whether EVs can shield the target tumors from DNMT3B inhibitors. Additionally, DNMT3B has been reported to contribute in chemotherapy resistance [37] and promotes radio-resistance in nasopharyngeal carcinoma [38]. DNMT3B in BAL EVs mirrors the expression in lung tumor tissue, and this might give a novel opportunity to use it as a marker for chemotherapy and radio-resistance. A second main functional category up-regulated in sEVs showing the same direction of regulation as in tumor tissue was NADP-binding which includes proteins interacting selectively and non-covalently with the oxidized form, NADP+, or the reduced form, NADPH. Evidence has been gathered showing that the alteration of NADPH metabolism is crucial for cell proliferation and survival. For example, the malic enzyme showed a four-fold change in tumor vs. control has been described to enhance the pentose phosphate pathway through direct binding and activating 6-phosphogluconate dehydrogenase, leading to tumor cell proliferation [39].

Integrins are bidirectional cell surface molecules are known to play a role in cancer cell survival and chemoresistance. In some tumors, higher integrin expression was associated with increased malignancy and metastasis formation. However, the down-regulation of integrins was also reported in cellular proliferation and invasion through epithelial-to-mesenchymal transition [40,41]. Herein, we demonstrate that ECM receptor interactions with its cellular constituents are downregulated in sEVs which may cause weak protein-protein interactions of adhesion proteins.

Macrophages (60%), lymphocytes (24%), and neutrophils (13%) represent the main cellular components in BAL [42]. The acellular BAL and BAL sEVs proteome mainly contained markers of B cell lineage. Several have reported the role of B cells in tumor immunity. For example, Bruno et al. observed an increased number of intratumoral B cells in freshly isolated primary tumors from NSCLC patients relative to B cells from tumor-adjacent tissues [43]. These tumor infiltrating lymphocytes B can generate anti-tumor antibodies [44], which in turn activate dendritic cells (DCs) to internalize tumor antigens and subsequently activate tumor-reactive T cells. Even though sEVs B cell markers expression was invariant between control and cancer, *IGLV3-1* was significantly upregulated, after correcting for multiple testing (>one order of magnitude), in sEVs BAL fraction from cancer patients compared to control (Appendix A). It would be interesting to explore the role of immunoglobulin-bound vesicles in tumor immune evasion. The binary complex could function as a pro-tumor effect, protecting cancer cells from the attack of cytotoxic T-cells.

The abundance scores for eight immune and two stromal cell markers in acellular BAL were randomly clustered in cancer versus non-cancer samples (Appendix A). For BAL sEVs, monocytic lineage, and fibroblasts marker scores weakly, which correlated with cancer and poor prognosis (Figure 6). The immune and two stromal cell markers’ abundance scores perfectly clustered the normal tumor tissues and xenotransplanted tumor tissue into distinct clusters (Appendix A). It is imperative to understand the tumor cell’s contribution to the sEVs population in comparison to the immune and stromal cells pool.

It will be necessary to deconvolute the contribution of each entity being, vesicles, soluble proteins, and cell populations (tumor, immune system, and stromal compartment) in the BAL fluid of lung cancer patients to provide clearer evidence of the role in tumor immune evasion that can be translated into a targeted treatment. The similarity in regulated KEGG functions in sEVs and tumor tissue appears promising in terms of exploring sEVs for diagnostic and prognostic markers in lung cancer. This along with a high protein coverage compared to other proposed fractionation methodologies of BAL sEVs constitute an interesting target in lung diseases in general.

## 4. Materials and Methods

### 4.1. Patient Samples

BAL samples were collected prospectively from patients with suspected lung cancer undergoing diagnostic bronchoscopy at the Unidade de Técnicas Invasivas Pneumológicas, Pneumologia II, Hospital Pulido Valente, Centro Hospitalar Lisboa Norte. Human Ethics approval was received from Centro Hospitalar Lisboa Norte and National Health Institute Dr. Ricardo Jorge Research Ethics Committees (DIRCLN-8ABR2014-130). All methods were performed in accordance with the relevant guidelines and regulations. All procedures performed in studies involving human participants were in accordance with the ethical standards of the institutional and national research committee and with the 1964 Helsinki Declaration and its later amendments or comparable ethical standards. Informed consent of all participants was obtained. A total number of 91 BAL samples were collected between April and July 2014, which was followed up two years later. Patient diagnosis was obtained by BAL cytology and whenever available biopsy followed by histology. Patient disease status was assessed at two different time points in August 2014 and June 2016. Samples (*n* = 24) used in the current study were selected to obtain 12 controls and 12 cancer patients. Non-lung cancer patients, referred to in this study as control, were diagnosed with diseases such as COPD, interstitial lung disease, bronchiectasis, heart failure, asthma, chronic cough, and pulmonary nodule follow-up. After the initial bronchoscopy, negative “Suspicious” patients were approached in the following way: a subset of the patients underwent CT-guided transthoracic needle biopsy, others repeated bronchoscopy and some of them were submitted to thoracic surgery. In the remaining patients presenting with small lung nodules, follow-up was performed according to Fleischner Society Guidelines [45]. The mean time length to the diagnosis of lung cancer was 100.61 days after the first bronchoscopy.

### 4.2. BAL Processing

In most cases, BAL was targeted toward affected lung segments. The procedure was performed by wedging the bronchoscope in a subsegmental bronchus. Usually, three lavages were performed using approximately 50 mL of 0.9% saline solution per lavage. The recovered fluid was placed at 4 °C immediately. BAL was centrifuged at 320× *g* for 10minutes at 4 °C to remove the cellular fraction. The resulting cell-free supernatant was immediately aliquoted and frozen at −80 °C until further analysis.

### 4.3. Isolation of Extracellular Vesicles from Human Acellular BAL

Frozen acellular BAL specimens were thawed and centrifuged at 3000× *g* for 20 min at 4 °C and then at 12,000× *g* for 60 min at 4 °C. Clarified plasma was ultracentrifuged in a Beckman Coulter Optima TM L-80XP Ultracentrifuge at 100,000× *g* at 4 °C for 120 min with a Type 50.2 Ti rotor (k-factor: 157.7) to pellet EVs. The supernatant was carefully removed, and crude EV-containing pellets were resuspended in ice-cold phosphate-buffered saline (PBS), followed by floatation on sucrose cushion (30%, D2O) for 60 min at 100,000× *g* at 4 °C to remove non EV protein complexes. After washing by pelleting the EVs collected in the sucrose cushion for 16h at 100,000× *g* at 10 °C, EVs were resuspended in PBS and subjected to NanoSight NS300 analysis (Malvern Instruments, Inc., Westborough, MA, USA).

### 4.4. Protein Measurements

Protein concentrations in isolated exosome fractions were measured using a BCA protein assay kit (Pierce Biotechnology, Rockford, IL, USA) according to the manufacturer’s instructions.

### 4.5. Transmission Electron Microscopy

TEM of isolated EVs was performed at the Instituto Gulbenkian de Ciência—Electron Microscopy Facility. Freshly isolated EVs were put on a copper-palladium 100 mesh grid coated with 1% formvar in chloroform and carbon. The grids were glow-discharged before the adhesion of samples. Samples were fixed with 2% formaldehyde in PBS, washed ten times with distilled water, and stained with 2% uranyl acetate in ddH2O. A Tecnai G2 Spirit BioTWIN from FEI operating at 120 keV and equipped with an Olympus-SIS Veleta CCD Camera was used for imaging.

### 4.6. Immunoblotting

BAL fluid and BAL derived-EVs proteins were quantified using Pierce™ BCA Protein Assay Kit (Thermo Scientific Pierce Biotechnology, P.O. Box 117, Rockford, USA). A constant protein amount of (10 µg) of BAL-EVs and BAL was separated on 10% SDS-PAGE under reducing conditions (4×-loading buffer: 8% SDS, 40% (*w*/*v*) glycerol, 20% Beta-mercaptoethanol, 0.08% bromophenol blue) and transferred to a Hybond PVDF membrane (Amersham, GE, Chicago, IL USA) or stained with Coomassie brilliant blue (Merck, Kenilworth, NJ, USA). The membrane was then incubated with primary antibody directed against CD63 (AB0047-200, SICGEN at a dilution of 1:2500) and HRP-conjugated secondary antibody donkey anti-goat (Jackson ImmunoResearch, Cambridge, UK). Signals were detected by enhanced chemiluminescence reagents ECL Western blot Detection (Amersham, GE, USA). Intensity of the bands was quantified using ImageLab Software. HeLa total cell lysate was analyzed in parallel using the same conditions as a positive control for anti-CD63. Images obtained were processed by the open-source software ImageJ (imagej.net) for quantitation.

### 4.7. Peptide Sample Preparation

Protein solutions containing SDS and DTT were loaded onto filtering columns and washed exhaustively with 8M urea in HEPES buffer [46]. Proteins were reduced with DTT and alkylated with IAA. Protein digestion was performed by overnight digestion with trypsin sequencing grade (Promega).

### 4.8. Mass Spectrometry Analysis

The peptides were analyzed on an Orbitrap Fusion Lumos Tribrid Mass Spectrometer (Thermo Fisher Scientific, San Jose, CA, 95134, USA) coupled with an EASY nLC1200 nanoflow liquid chromatography system (Thermo Fisher Scientific). The peptides reconstituted in 0.5% FA (formic acid) were loaded on a trap column (Acclaim™ PepMap™ 100, LC C18, 5 μm, 100 μm × 2 cm, nanoViper, Thermo Fisher Scientific) at a flow rate of 4 μL/min. The peptides were resolved at 0.3 μL/min flow rate using an increasing gradient of solvent B (0.1% FA in 90% ACN) on an analytical column (Easy-Spray™ PepMap™ RSLC C18, 2 μm, 75 μm × 50 cm, Thermo Fisher Scientific), which was fitted with an EASY-Spray ion source that was operated at a voltage of about 2.1 kV. The total run time was 120 min. Mass spectrometry analysis was carried out in a DDA (data-dependent acquisition) with a full scan in the *m*/*z* (mass-to-charge ratio) range of 350 to 1800 in the “Top Speed” setting, 3 sec per cycle. MS1 and MS2 were acquired for the precursor ions and the peptide fragmentation ions, respectively. MS1 scans were measured at a resolution of 120,000 at an *m*/*z* of 200. MS2 scans were acquired by fragmenting precursor ions using the HCD (higher-energy collisional dissociation) method, which was set to 32% of collision energy with 5% of stepped collision energy, and detected at a mass resolution of 30,000 at an *m*/*z* of 200. AGC (automatic gain control) targets were set to one million ions for MS1 and 0.05 million ions for MS2. The maximum ion injection time was set to 50 ms for MS1 and 100 ms for MS2. The precursor isolation window was set to 1.6 *m*/*z* with 0.4 *m*/*z* of offset. Dynamic exclusion was set to 35 sec, and single charged ions were rejected. Internal calibration was carried out using the lock mass option (*m*/*z* 445.12002) from ambient air.

### 4.9. Database Search

The obtained data from the 24 LC-MS runs were searched using VEMS [47] and MaxQuant [48]. The MSMS spectra were searched against a standard human proteome database from UniProt (3AUP000005640). Permutated protein sequences, where Arg and Lys were not permutated, were included in the database for VEMS search. Trypsin cleavage allowing a maximum of 4 missed cleavages was used. Carbamidomethyl cysteine was included as a fixed modification. Methionine oxidation and N-terminal protein acetylation were included as variable modifications. 5 ppm mass accuracy was specified for precursor ions and 10 *m*/*z* for fragment ions. The false discovery rate (FDR) for protein identification was set to 1% for peptide and protein identifications. No restriction was applied for minimal peptide length for VEMS search. The identified proteins were divided into evidence groups as defined by Matthiesen et al. [49].

### 4.10. Functional Analysis of Identified Proteins in BAL Fractions

All the protein identifications by VEMS of each of the fractions BAL, depleted BAL, lEVs, and sEVs were subjected to KEGG functional analysis using the hypergeometric function as described previously [50]. The heatmap was plotted with the R package ComplexHeatmap [51].

### 4.11. Quantitative Analysis

The quantitative comparison of acellular BAL, BAL exosomes, patient match normal versus tumor tissue, and patient-matched normal versus mouse xenotransplant tissue were based on MaxQuant [48] analysis using the same parameter settings. Because of the difference in chromatography, each data set was analyzed separately and iBAQ values combined in R for further statistical analysis. IBAQ were preprocessed by three approaches: (1) removing common MS contaminants followed by log_2_(x + 1) transformation, (2) removing common MS contaminants followed by log_2_(x + 1) transformation and quantile normalization, and (3) removing common MS contaminants followed by log_2_(x + 1) transformation, quantile normalization and abundance filtering to optimize overall Gaussian distribution of all quantitative values. The top regulated proteins based on the different strategies were similar but with a small difference in the ranking of the proteins. The quantile normalized data were used for subsequent complete functional regulation analysis.

### 4.12. Complete Functional Regulation Analysis

The quantitative data for acellular BAL, BAL exosomes, patient match normal versus tumor tissue, and patient-matched normal versus mouse xenotransplant tissue were subjected to completed functional regulation analysis as previously described [21]. The quantitative data was matched against the functional categories in KEGG [52], gene ontology [53], HPRD [54], CORUM [55], and MSigDB [56]. Only significantly enriched functional categories with an enrichment *p* value < 0.05, a regulation higher than two-fold, and *p* value on regulation < 0.05 were maintained. The direction of the regulation of the functional categories across different sample comparisons was compared in a heatmap (Figure 5).

## 5. Conclusions

To our knowledge, we provide the first MS-based proteomic comparison of BAL fractions obtained by sequential centrifugation. We conclude that the proteome of sEVs displays the highest protein complexity of all the BAL fractions investigated. Our results prompt further studies in larger cohorts to clarify if BAL sEV proteomes can provide practical clinical information on lung cancer staging. Finally, the results highlight for the first time the role of EGFR downstream effectors in sEVs with potential implications in drug resistance.

## Figures and Tables

**Figure 1 cancers-12-03450-f001:**
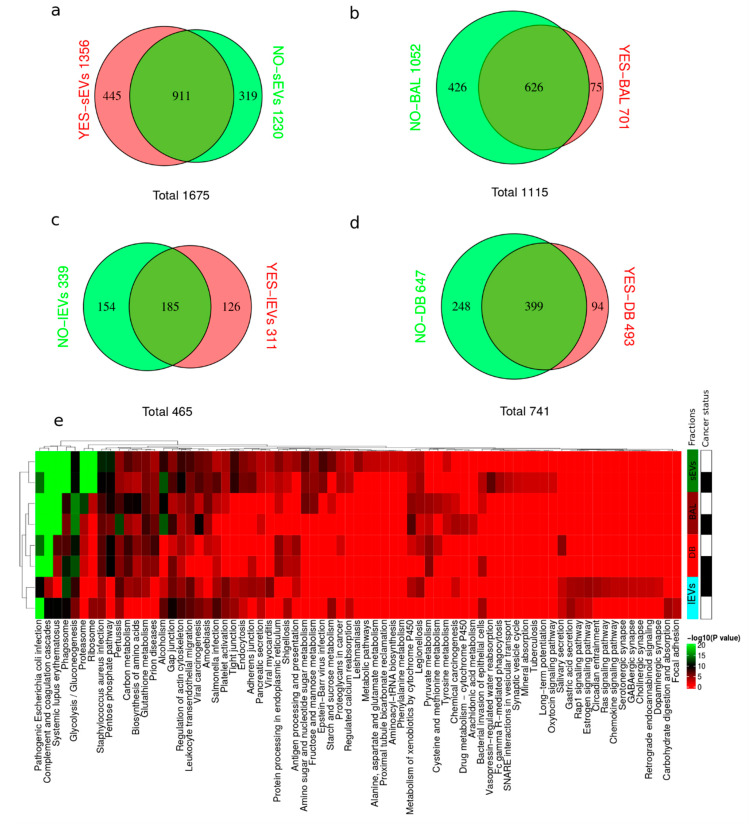
Proteome overview in bronchoalveolar lavage (BAL) fractions. Venn diagrams comparing cancer (YES) versus control (NO) in: (**a**) sEVs, (**b**) BAL, (**c**) lEVs, and (**d**) DB; (**e**) Heat-map of KEGG pathway enrichment analysis. Functional enrichment analysis was performed using R. *p*-value of each KEGG pathway was <0.05. The shades of the colors reflected the –log10 (*p*-value) of the enrichment analysis of proteins identified in different subcellular BAL fractions for NO (white) and YES (black) cancer status samples. Large extracellular vesicles (lEVs), BAL vesicle-depleted (DB), acellular BAL (BAL), small extracellular vesicles (sEVs).

**Figure 2 cancers-12-03450-f002:**
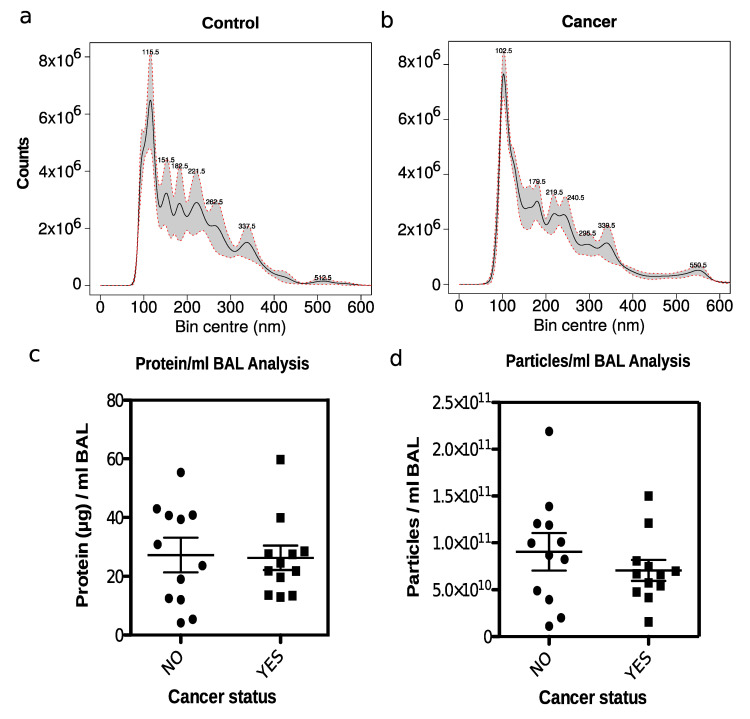
Nanoparticle tracking analysis of sEV samples: (**a**) particle counts versus particle size in nm for a control sample; (**b**) and a lung cancer sample; (**c**) protein concentration in sEVs versus cancer status; (**d**) particle concentration versus cancer status.

**Figure 3 cancers-12-03450-f003:**
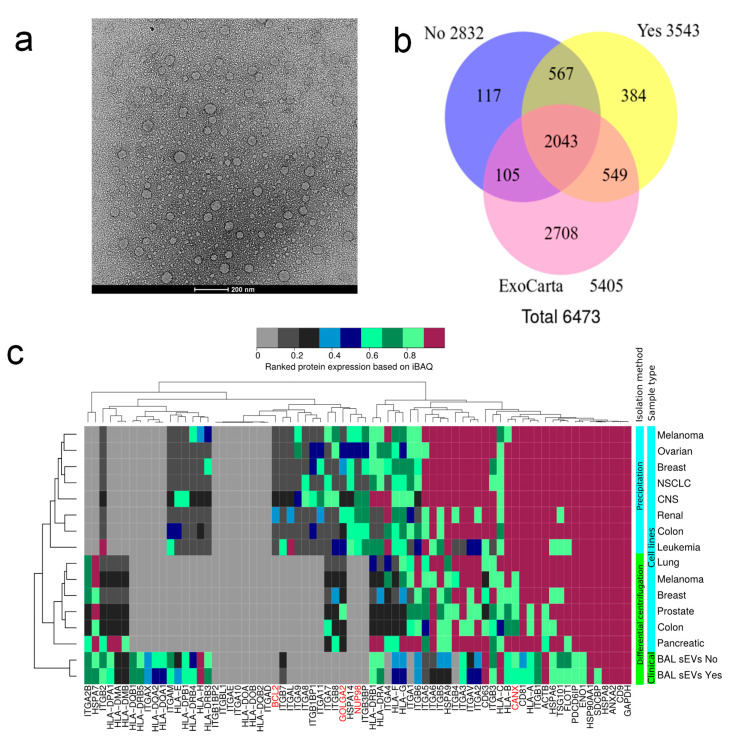
Extracellular vehicle (EV) quality assessment: (**a**) representative transmission electron microscopy images of enriched sEVs; (**b**) Venn diagram indicating overall protein identification overlap with ExoCarta database; (**c**) MS-based quantitative comparison between enriched BAL sEVs and cell line isolated EVs of frequently reported exosome protein markers and 10 most abundant exosome marker from ExoCarta. Red labeled proteins indicate non EV proteins.

**Figure 4 cancers-12-03450-f004:**
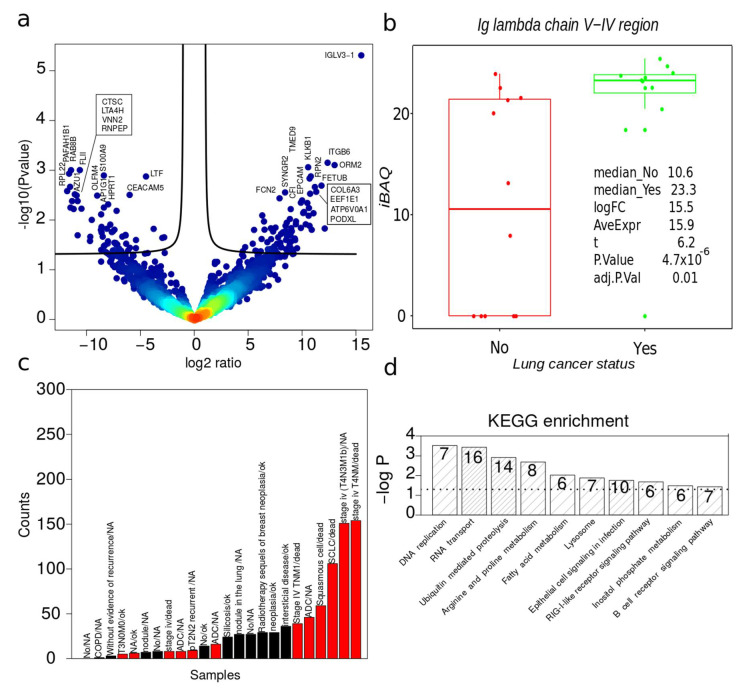
Dysregulated lung cancer proteins in BAL EVs: (**a**) Volcano plot summarizing the main dysregulated proteins where –log10 *p* values were corrected for gender and smoking; (**b**) top regulated protein after correction for multiple testing; (**c**) number of unique proteins per sample (red bars indicate lung cancer and black bars represent controls); (**d**) KEGG functional enrichment of cancer unique proteins. The numbers in the bars represent the number of cancer unique proteins in the functional category. The dashed line indicates the *p*-value threshold (<0.05).

**Figure 5 cancers-12-03450-f005:**
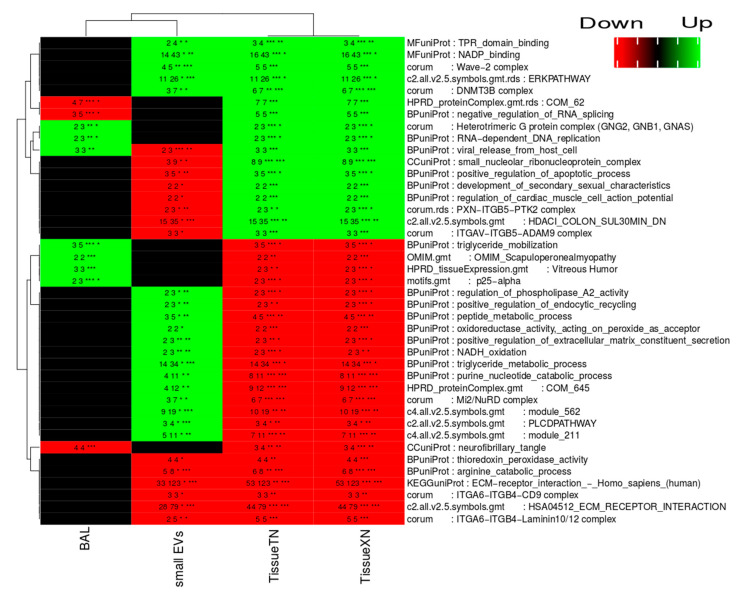
Complete functional regulation and enrichment analysis of proteins identified in acellular BAL (BAL), small EVs, lung tumor tissue (TissueTN), and xenotransplant lung tumor tissue (TissueXN). The color code in the heatmap reflects the regulation level (black unregulated, green significant up-regulated in tumor, and red significantly down-regulated in tumor). The text in the cells indicates the number of proteins identified in each functional category, the number of proteins annotated in the given functional category and the significance level of regulation, and significance level of enrichment (* *p* <0.05, ** *p* < 0.01, and *** *p* < 0.001).

**Figure 6 cancers-12-03450-f006:**
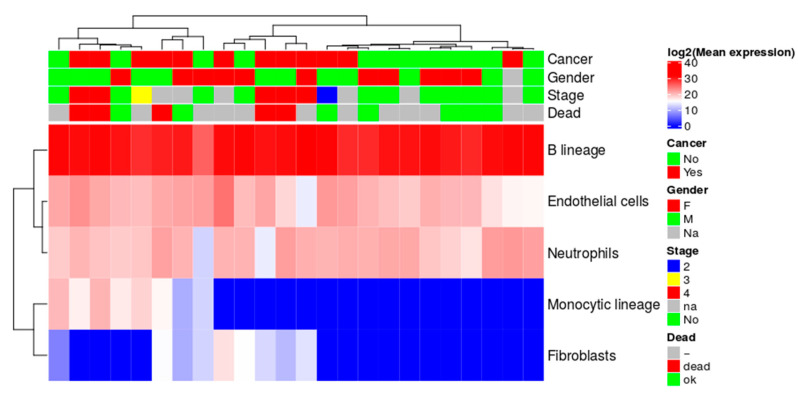
Average abundance scores of protein markers for eight different immune and two stromal cell populations in sEVs BAL samples.

**Table 1 cancers-12-03450-t001:** Baseline characteristic of the clinical samples.

	NO(*N* = 12)	YES (*N* = 12)	Total (*N* = 24)	*p* Value
Age				0.355
<55	4 (33.3%)	1 (8.3%)	5 (12.5%)	
>55	1 (8.3%)	1 (8.3%)	2 (8.3%)	
NA	7 (58.3%)	10 (83.3%)	17 (70.8%)	
Smoking History				0.370
Current smoker	4 (33.3%)	3 (25.0%)	7 (29.2%)	
Former smoker	2 (16.7%)	6 (50.0%)	8 (33.3%)	
Nonsmoker	2 (16.7%)	1 (8.3%)	3 (12.5%)	
Unknown	4 (33.3%)	2 (16.7%)	6 (25.0%)	
Gender				0.100
NA	0 (0.0%)	1 (8.3%)	1 (4.2%)	
F	8 (66.7%)	3 (25.0%)	11 (45.8%)	
M	4 (33.3%)	8 (66.7%)	12 (50.0%)	
Batch				0.390
1	2 (16.7%)	5 (41.7%)	7 (29.2%)	
2	9 (75.0%)	6 (50.0%)	15 (62.5%)	
3	1 (8.3%)	1 (8.3%)	2 (8.3%)	
Stage				<0.001
2	0 (0.0%)	1 (8.3%)	1 (4.2%)	
3	0 (0.0%)	1 (8.3%)	1 (4.2%)	
4	0 (0.0%)	5 (41.7%)	5 (20.8%)	
NA	1 (8.3%)	5 (41.7%)	6 (25.0%)	
No	11 (91.7%)	0 (0.0%)	11 (45.8%)	
Status				0.012
Alive	12 (100.0%)	7 (58.3%)	19 (79.2%)	
Dead	0 (0.0%)	5 (41.7%)	5 (20.8%)

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
