# Peer review of "Is the Proteome of Bronchoalveolar Lavage Extracellular Vesicles a Marker of Advanced Lung Cancer?"

_cancers, 2020, doi:10.3390/cancers12113450_

Round 1

Reviewer 1 Report

Protein complexity in bronchoalveolar lavage extracellular vesicles correlates with stage four cancer and short term mortality.  [cancers-984646]

The authors report a study based on LC-MS proteomic profiling of extracellular vesicles derived from bronchoalveolar lavage of lung cancer patients and corresponding apparent cancer-free controls.  The findings are largely descriptive and while bioinformatic assessments are employed to characterize the resulting data, the narrative of the manuscript does not progress towards a central finding or insight.  The authors state that the “study provides insights into the proteome content of BAL EVs and its correlation to clinical outcome,” however as the clinical samples are from predominantly advanced stage lung cancer cases, the clinical outcome would be expected to be poor and it is not surprising that the BAL EVs would manifest distinct protein cargoes compared to the controls.  The manuscript does have merit as a catalogue of protein features conveyed by EVs in the context of cancer, nevertheless the biomarker utility of the findings is somewhat more elusive.

There are some technical issues that should be addressed as well.  Based on the protein content and particle counts of the EV isolates, the particle to protein ratio is approximately 4e9 particles/µg protein.  This is indicative of relatively low purity according to the measure suggested by Webber and Clayton [doi: 10.3402/jev.v2i0.19861].

From the heat-map in Figure 1, the –log10(P value) reported for the various KEGG pathways does seem to vary to some extent between the l/sEVs, depleted BAL and unfractionated BAL, but differences between case/control is less pronounced, suggesting that many of these enriched pathways are not cancer related.  It would be more informative to specifically report pathways that are differentially manifest in the cancer BAL EVs. The cancer status legend suggests the case/control order is reversed relative to the other BAL fractions, confirm that this is the case.

It is less than ideal that the stage status is unknown for 5/12 of the case samples.

The nanotracking data presented in Figure 2 does indeed indicate presence of sEV/exosome-sized particles; however, the distribution is relatively broad and does suggest some issue with the purity of the isolates.

The information presented in Figure 4b is informative, but less interesting compared to the protein features that were found to be significantly differential between cancer positive and negative BAL EVs.  A volcano plot that reports these proteins with the most prominent features labeled would enhance the impact of the figure.

The functional regulation analyses comparing unfractionated BAL, BAL sEVs and tumor tissues that is reported in Figure 5 provides intriguing evidence that lung tumor tissue features are conveyed by EVs.  It is rather information-dense and further explication in the Discussion could be helpful. TissueXN is assumed as xenograft – perhaps define (along with TissueTN) in the legend.

The BAL sEV immune cell signatures indicated in Figure 6 suggest capacity for disseminated sEVs to report on immune status of the tumor microenvironment, which is an interesting finding.  Nevertheless, the reporting of the data in the Results and Discussion is somewhat cursory.  Expanding this would improve the impact of the manuscript.

The title does little to kindle the interest of the (potential) reader, nor does it summarize the essential findings of the paper.

Author Response

We thank the reviewers for constructive criticisms which have helped us to improve the manuscript.

Reviewer 1:

The authors report a study based on LC-MS proteomic profiling of extracellular vesicles derived from bronchoalveolar lavage of lung cancer patients and corresponding apparent cancer-free controls. The findings are largely descriptive and while bioinformatic assessments are employed to characterize the resulting data, the narrative of the manuscript does not progress towards a central finding or insight. The authors state that the “study provides insights into the proteome content of BAL EVs and its correlation to clinical outcome,” however as the clinical samples are from predominantly advanced stage lung cancer cases, the clinical outcome would be expected to be poor and it is not surprising that the BAL EVs would manifest distinct protein cargoes compared to the controls. It is less than ideal that the stage status is unknown for 5/12 of the case samples. The title does little to kindle the interest of the (potential) reader, nor does it summarize the essential findings of the paper.

Authors:

We agree that the clinical samples has limitations concerning staging and total cohort size. We therefore in agreement with the reviewer 1’s comment changed the title to “Is the proteome of bronchoalveolar lavage extracellular vesicles a marker of advanced lung cancer?”. Nevertheless an statistical significant association with protein complexity was found which was also confirmed when exploring data on lung tissue and PDX models (figure 4B and figure S6B). Furthermore, the almost perfect separation of BAL versus controls in the PCA plot clearly suggests that the EV proteomes contains variation related to cancer (Figure S6). In addition, functional enrichment analysis on cancer unique proteins in figure 4d resulted in KEGG pathways known to be associated with cancer. This is the first description of BAL EVs proteome of lung cancer suspects and it has not been previously addressed whether protein expression of EVs isolated from BAL correlate with lung cancer diagnostic.

Reviewer:

The manuscript does have merit as a catalogue of protein features conveyed by EVs in the context of cancer, nevertheless the biomarker utility of the findings is somewhat more elusive.

There are some technical issues that should be addressed as well.  Based on the protein content and particle counts of the EV isolates, the particle to protein ratio is approximately 4e9 particles/µg protein. This is indicative of relatively low purity according to the measure suggested by Webber and Clayton [doi: 10.3402/jev.v2i0.19861]. The nanotracking data presented in Figure 2 does indeed indicate presence of sEV/exosome-sized particles; however, the distribution is relatively broad and does suggest some issue with the purity of the isolates.

Authors:

We agree that a better discussion of the quality of the EVs are required, and is also in line with reviewer 2’s request. However, we are unconvinced with Webber and Clayton [doi: 10.3402/jev.v2i0.19861] argument to use protein/particles to assess purity of EVs preparations. Nevertheless, we now discuss our observation objectively and compare with Webber and Clayton in the discussion section. We know from flow cytometry that our samples obtained with the described methodology typically contain >90% of vesicles. In addition, we accessed the level of marker proteins from other cellular organelles based on request from reviewer 2. In figure 3c, we observe that the level of BCL2 (mitochondrial marker), GOLGA2 (golgi marker), NUP98 (nuclear marker) and CANX (endoplasmatic reticulum marker) is much lower in our clinical EV isolates than what was obtained from cell lines from other studies. Still, the clinical EV isolates displayed similar levels compared to cell line EVs of commonly used EV markers.

As described in Webber and Clayton ratios >3x1010 particles/microgram of proteins equate to high vesicular purity, ratios of 2x109 to 2x1010 particles/microgram of protein represent low purity. However as shown in Fig 3 of Webber and Clayton high vesicular purity is achievable solely from cell culture conditioned media. When comparing purity ratio between conditioned media and two body fluids (urine and serum), the authors report a 10-fold lower ratio by pelleting and wash. The BAL EV isolates presented a particle to protein ratio of approximately 4x109 particles/µg protein which is in the range of the ratios obtained for urine and serum in Webber and Clayton. In our study the isolation method used included pelleting+sucrose cushion+ wash which was shown to increase EVs purity 1.7-fold.

Reviewer:

From the heat-map in Figure 1, the –log10(P value) reported for the various KEGG pathways does seem to vary to some extent between the l/sEVs, depleted BAL and unfractionated BAL, but differences between case/control is less pronounced, suggesting that many of these enriched pathways are not cancer related. It would be more informative to specifically report pathways that are differentially manifest in the cancer BAL EVs. The cancer status legend suggests the case/control order is reversed relative to the other BAL fractions, confirm that this is the case.

Authors:

We thank the reviewer for this comment, we agree that the difference between control and cancer is also relevant and the main point to illustrate at this point of the manuscript. We also agree that our argumentation for further exploration of sEVs in the previous manuscript was not well justified. We have consequently created an additional figure to clarify this issue. In addition, we see observe several functional categories related to cancer almost exclusively in fractions BAL EVs. For example, proteasome, ribosome and tight junctions. Finally, the reversal for case/control order is caused by the clustering of the rows. The samples are clustered correctly based on the functional annotation, strongly suggesting that each fraction contains distinct protein functional groups.

Reviewer:

The information presented in Figure 4b is informative, but less interesting compared to the protein features that were found to be significantly differential between cancer positive and negative BAL EVs. A volcano plot that reports these proteins with the most prominent features labeled would enhance the impact of the figure.

Authors:

We thank for this comment and agree that a volcano plot at this point will aid the reader to quickly overview the main regulated proteins. We have consequently added a volcano plot in figure 4b. In addition, Table S1 provides a list of all proteins identified in BAL sEVs, and respective statistical parameters such as mean, coefficient of variance, fold change, P-value, adjusted P-value, B-statistic value and the number of samples in which the protein has been identified.

Reviewer:

The functional regulation analyses comparing unfractionated BAL, BAL sEVs and tumor tissues that is reported in Figure 5 provides intriguing evidence that lung tumor tissue features are conveyed by EVs. It is rather information-dense and further explication in the Discussion could be helpful. TissueXN is assumed as xenograft – perhaps define (along with TissueTN) in the legend.

Authors:

We agree with reviewer. We have followed the reviewers suggestion and updated the legend accordingly and elaborated the discussion.

The BAL sEV immune cell signatures indicated in Figure 6 suggest capacity for disseminated sEVs to report on immune status of the tumor microenvironment, which is an interesting finding.  Nevertheless, the reporting of the data in the Results and Discussion is somewhat cursory.  Expanding this would improve the impact of the manuscript.

Authors:

Again a useful comment for manuscript improvement, we have updated the result and discussion to discuss the possible impact of immune related factors observed in EVs.

Reviewer 2 Report

Major point:

1.To my knowledge, no EVs isolation protocol allows the recovery of a pure EVs subpopulation, and the majority of the available protocols have a profound influence on the omics results applied post-isolation. Because of that, proteomic profiles of EVs have been highly dependent on your isolation protocol. Therefore, authors need to include in EVs studies all the detailed information to define extracellular vesicles and their function. What control did they use to rule out contamination?

2. Authors mention about efforts to elucidate the role of EVs in drug resistance implicated in downstream effectors of EGFR signaling cascade involving RAS-RAF-MEK-ERK pathways are needed. It is unclear how they believe those MS data reflect the role of EGFR downstream effector in sEVs. All EGFR associated pathways and downstream signaling pathways need to be confirmed.

3. Immunoblotting should be used to confirm some of the essential protein makers correlate with stages four cancer, not just based on MS data.

Author Response

We thank the reviewers for constructive criticisms which have helped us to improve the manuscript.

Reviewer 2:

1.To my knowledge, no EVs isolation protocol allows the recovery of a pure EVs subpopulation, and the majority of the available protocols have a profound influence on the omics results applied post-isolation. Because of that, proteomic profiles of EVs have been highly dependent on your isolation protocol. Therefore, authors need to include in EVs studies all the detailed information to define extracellular vesicles and their function. What control did they use to rule out contamination?

Authors:

This is a good comment. We agree with the reviewer and we have updated the EV quality control tool to also asses the level of protein markers considered contaminants. In figure 3c, we observe that the level of BCL2 (mitochondrial marker), GALGA2 (golgi marker), NUP98 (nuclear marker) and CANX (endoplasmatic reticulum marker) is much lower in our clinical EVs isolate than what was obtained from cell lines. Please also refer to our answer to reviewer one on this issue.

2. Authors mention about efforts to elucidate the role of EVs in drug resistance implicated in downstream effectors of EGFR signaling cascade involving RAS-RAF-MEK-ERK pathways are needed. It is unclear how they believe those MS data reflect the role of EGFR downstream effector in sEVs. All EGFR associated pathways and downstream signaling pathways need to be confirmed.

Authors:

It has previously been demonstrated that CD20 targets, excessively secreted in EVs, shield the CD20 cellular target on cancer cells [33]. Since we observe EGFR in EVs this may influence drugs targeting EGFR in a similar way. We agree with the authors that we do not provide proof of this mechanisms but rather discus it as a interesting topic for future studies in the discussion.

3. Immunoblotting should be used to confirm some of the essential protein makers correlate with stages four cancer, not just based on MS data.

AU: The EVs protocol gives very limited amount EVs and we therefore do not have sufficient material to perform extensive Western blots. We consequently focused on using the isolated EVs for MS characterization and what was left was used to test for exosome marker CD63. This is a limitation of the use of EVs for proteomics studies that is not frequently mentioned.

Reviewer 3 Report

The authors presents a case-control study of 24 bronchoalveolar lavage (BAL) extracellular vesicles (EV) sample. They found that BAL EVs proteome complexity correlated with lung cancer stage 4 and mortality within two years follow-up.

        Specific points:

  • please define LC-MS
  • Please indicate the R package used for the heatmap of the figure 1.
  • Figure 1 indicates 4 fractions: IEVs, DB, BAL and sEVs. In the main text section 2.2 line 109-110 are indicated 3 fractions. Please explain better. I lost DB.
  • Figure 3 b is not commented/cited in the main text. Please add

Author Response

Authors:

We thank reviewer 3 for the contructive comments which improved the clarity of the manuscript content.

Reviewer:

please define LC-MS

AU: We defined LC-MS in abstract and first occurrence in main text.

    •  

Please indicate the R package used for the heatmap of the figure 1.

AU: We added: “The heatmap was plotted with the R package ComplexHeatmap [48].”

    •  

Figure 1 indicates 4 fractions: IEVs, DB, BAL and sEVs. In the main text section 2.2 line 109-110 are indicated 3 fractions. Please explain better. I lost DB.

AU: We modified the protocol outline Figure S1 and amended the following statement to clarify: “BAL fluid (acellular BAL, aBAL) from two different cancer status samples (NO and YES) were fractionated by sequential centrifugation and ultracentrifugation into three fractions: 1) BAL vesicle-depleted BAL (DB), 2) lEVs and 3) sEVs (Figure S1).”

    •  

Figure 3 b is not commented/cited in the main text. Please add

AU: We inserted a reference to figure 3b:

All identified proteins from controls and lung cancer in sEV were compared to proteins deposited in the exosome database ExoCarta, showing an overlap > 70% for the two different cancer status samples (Figure 3b).

Round 2

Reviewer 2 Report

The authors have addressed all my concerns and therefore I support publication without further changes.